# Impact of Antihypertensive Treatment Adherence on Premature Mortality over Seven Years: A Follow-Up Investigation

**DOI:** 10.3390/jcm14238321

**Published:** 2025-11-23

**Authors:** Nafisa Mhna Kmbo Elehamer, Mohammed Merzah, Sami Najmaddin Saeed, János Sándor, Árpád Czifra

**Affiliations:** 1Department of Public Health and Epidemiology, Faculty of Medicine, University of Debrecen, H-4028 Debrecen, Hungary; elehamer.nafisa@med.unideb.hu (N.M.K.E.); mohammed.merzah@med.unideb.hu (M.M.); saeed.sami@med.unideb.hu (S.N.S.); czifra.arpad@med.unideb.hu (Á.C.); 2Department of Health Education, Faculty of Public and Environmental Health, University of Khartoum, Khartoum 11115, Sudan; 3Doctoral School of Health Sciences, University of Debrecen, H-4032 Debrecen, Hungary; 4HUN-REN-DE Public Health Research Group, Department of Public Health and Epidemiology, Faculty of Medicine, University of Debrecen, H-4012 Debrecen, Hungary; 5Department of Community Health Techniques, Polytechnic College- Karbala, Al-Furat Al-Awsat Technical University, Kufa 54001, Iraq

**Keywords:** hypertension, lethal prognosis, risk factors, treatment adherence, cohort

## Abstract

**Background/Objectives:** Despite the availability of highly effective medications, hypertension is among the most important risk factors for mortality. Because medication adherence is challenging worldwide, enhancing it to improve the prognosis of hypertension is useful. The aim of this study was to describe the prevalence of antihypertensive medication nonadherence among individuals aged 18–64 years in a deprived Hungarian population and its determinant factors, and to quantify the impact of antihypertensive medication nonadherence on premature mortality. **Methods:** We used data from a cohort of hypertensive individuals aged 18–64 years linked to the Health Insurance Fund’s medication purchasing data. The antihypertensive treatment adherence appropriateness (ATAP) was computed as the ratio of the observed time when a patient was properly treated to their observed survival time. ATAP was dichotomized by an observed mean of 0.872. Using adjusted odds ratios (AORs) from multivariate logistic regression models with 95% confidence intervals (CIs), we analyzed the factors influencing the mortality risk in 4962 participants over seven years of follow-up. **Results:** A total of 493 deaths occurred. An extremely high mortality risk was observed among patients with inappropriate adherence (AOR = 56.2, 95%CI: 41.9–75.4), which could be attributed partly to residual confounding. Significant protective factors were female sex and high education attainment. However, older age and all investigated comorbidities (diabetes mellitus, ischemic heart disease, chronic obstructive pulmonary disease, and cancer) were significantly associated with an increased risk of a lethal outcome. Similarly, smoking was also a risk factor. **Conclusions:** Our investigation revealed the following: (1) in the studied group of patients aged 18–64 years from an extremely disadvantaged Hungarian population, 87.2% of the person-time was covered by the appropriate redemption of medications; (2) nonadherence to medication was more common among younger adults, men, Roma people, current smokers, and COPD patients, whereas the likelihood of appropriate adherence was higher among patients with diabetes mellitus; (3) medication nonadherence was an extremely strong risk factor for a lethal outcome of HTN during the 7-year follow-up period; and (4) methods by which nonadherent patient behavior can be detected should be applied rigorously, and the detected nonadherence should be considered a signal for intervention to improve the prognosis of HTN.

## 1. Introduction

Hypertension (HTN) or high blood pressure (BP) is a critical medical condition that can increase the risk of brain, heart, kidney, and other diseases [1,2,3]. It is considered a serious public health concern worldwide and a notable cause of premature death, making it a leading cause of health loss, with up to one in four men and one in five women experiencing this condition [1]. Among people aged 30–79 years, approximately 10 million deaths per year can be attributed to HTN [4,5,6]. Unfortunately, in many patients, HTN is not properly treated or controlled [1]. Approximately 50–70% of individuals diagnosed with HTN do not redeem antihypertensive medication (AHM) per prescription [7], and improper antihypertensive adherence (AH) leads to considerable disease progression, which increases cardiovascular complications and decreases therapeutic effectiveness [8,9].

In Europe, despite the 30–45% prevalence of HTN, which increases considerably with age, less than half of hypertensive individuals achieve the recommended BP targets, leading to high morbidity and mortality and considerable health care costs [10]. In contrast, Hungary has a higher prevalence of HTN than the rest of Europe, and HTN screening has revealed high rates of uncontrolled BP [1,11]. According to a World Health Organization report, the prevalence of HTN in Hungarian adults aged 30–79 years is above the average global prevalence of 48%, and the prevalence of uncontrolled HTN in Hungary is among the highest in Europe [1,12]. Although cardiovascular mortality is the main component of avoidable causes of death among adults under 75 years of age [13], we do not have epidemiological data on the extent to which individual risk factors for cardiovascular mortality in this age group contribute to the very high avoidable mortality in Hungary. The prognostic factors of HTN have been well established through comprehensive clinical research. Given this comprehensive understanding, the effective use of available therapeutic interventions is essential for optimizing patient outcomes [14]. Maintenance of high medication adherence is among the most important prognostic factors [15,16].

Many interventions have been shown to be effective at improving medication adherence. These include dose modification and a reduction in the number of medications, timely reminders and improved scheduling, patient education, expansion of the role of pharmacists, improved patient–provider communication, better descriptions of illness and medication, patient follow-up with support and motivation, and timely responses to patient feedback [17,18,19,20,21,22,23].

Despite the extensively documented importance of treatment adherence in HTN management, global adherence rates remain suboptimal. Given the persistent gap between existing adherence levels and optimal therapeutic outcomes, there is a pressing need for structured, organized, and multidisciplinary approaches [24,25,26,27,28]. Recent Hungarian investigations have also revealed considerable challenges related to primary medication adherence (59.4% of the cardiovascular prescriptions are dispensed [29]), and to the persistent use of antihypertensive medication for single-pill combinations; free equivalent dose combinations are 82–85% and 58–73%, respectively [30,31]. The importance of medication nonadherence in determining the poor Hungarian premature cardiovascular mortality has not yet been thoroughly investigated.

Our study was based on a cohort established by an interventional study and took advantage of the cohort members’ administrative records being traceable in the administrative database of the National Health Insurance Fund (NHIF). It aimed to (1) describe the prevalence of antihypertensive medication nonadherence among individuals aged 18–64 years in Hungary, (2) explore the determinant factors, (3) quantify the impact of antihypertensive medication nonadherence on premature mortality, and (4) evaluate the adequacy of methods used to control medication nonadherence as part of primary hypertension care.

## 2. Materials and Methods

### 2.1. Study Design and Setting

This was a cohort study. The cohort was established by a regional health examination survey (HES) conducted from November 2013 to June 2016, which was focused on one of the most deprived regions of Hungary, with the goal of reaching all adults aged ≥ 18 years in the targeted areas. Trained health professionals and nurses collected data, which included sociodemographic factors, comorbidities, lifestyle, and health status. Participants’ health insurance identifiers were also recorded [32,33,34,35]. Further details of the survey have been published elsewhere [34].

Records from the survey (N = 19,789) were linked to the administrative database of the NHIF using individual health insurance identifiers. Participants whose records included mistyped health insurance identifiers (N = 190), who had incorrect HES data (N = 9), and who did not reside in the intervention area (N = 550) were excluded. Because follow-up was possible for the period of 2016–2022 in the NHIF databases, the investigated sample was restricted to records from the HES conducted in 2014 and 2015. The original sample was restricted to individuals with hypertension detected at the HES who were aged 18–64 years. Participants with incomplete records were excluded from our analysis. A total of 4962 records were used in the statistical analyses (Figure 1). These participants were followed up in the NHIF administrative database for 7 years.

### 2.2. Variables

Patients’ sex (male and female), age group (18–45 and 46–64 years), educational attainment (≤8 years of schooling, vocational school, graduation, and advanced), self-declared ethnicity (Roma and non-Roma), and smoking status (nonsmoker, current smoker, and former smoker) were recorded at the HES.

BP was measured at the HES and was recorded as the mean of two SBP and DBP measurements. Patients were considered hypertensive if their SBP was ≥140 mmHg and their DBP was ≥90 mmHg, in accordance with the European Heart Association guidelines [36].

Comorbidities during the follow-up period determined on the basis of the NHIF’s drug consumption and hospital discharge records included diabetes mellitus (DM), ischemic heart disease (IHD), chronic obstructive pulmonary disease (COPD), and cancer (CA) [33]. The date of death during the follow-up for lethal outcomes was also determined using the NHIF database.

The antihypertensive treatment adherence appropriateness (ATAP, a medication possession ratio adapted to the structure of the available data) was computed for each participant using the NHIF’s drug consumption database. The number of years a person was treated during the follow-up was counted (treatment-years). The treatment status for each year was recorded as a binary value. A person was considered treated if they redeemed antihypertensive medications four times a year continuously because general medical practitioners (GPs) are allowed to prescribe antihypertensive drugs every three months. The survival time for each participant was determined as the number of years from the HES to either the year of death or the end of the follow-up period for those still alive (survival duration). The ratio of the treatment-years to the survival duration indicates the proportion of survival time during which treatment was administered, which offers a measure of treatment exposure relative to follow-up length. The mean ATAP was 0.872. ATAP was categorized as below or equal to the mean (treated inappropriately) or above the mean (treated appropriately).

### 2.3. Data Analysis

Descriptive statistics were used to determine the frequencies and crude percentages of explanatory variables for patients treated appropriately and for patients treated inappropriately. The chi-square test was applied to evaluate variations in proportions across groups.

A multivariable logistic regression model was used to determine the risk factors for a lethal outcome. ATAP, sex, age, level of education, Roma ethnicity, smoking status, and comorbidities (DM, IHD, COPD, and CA) were the tested explanatory variables. Adjusted odds ratios (AORs) with 95% confidence intervals were computed.

The data were analyzed using the IBM Statistical Package for Social Sciences (SPSS) version 28.0 (Armonk, NY, USA).

### 2.4. Ethical Aspects

The Ethics Committee of the Hungarian National Scientific Council on Health approved the study protocol (approval code: TUKEB 16676-3/2016/EKU, 0361/16; approval date: 30 March 2016). Prior to the commencement of HES data collection, all participants provided written informed consent to participate in the HES and allowed the storage and analysis of their data. The data used in this study were deidentified to protect participant privacy, and all data analyses were conducted in accordance with the ethical principles of the Declaration of Helsinki. Patients’ health insurance numbers were processed exclusively on computers within the National Institute of Health Insurance Fund by designated staff members who routinely handled sensitive data as part of their regular work duties. No other study personnel were permitted to access these data [32,33,34,35].

## 3. Results

Our study comprised 4962 individuals with HTN. A total of 78.2% (N *=* 3881) demonstrated appropriate treatment adherence, and 21.8% (N *=* 1081) had inappropriate treatment adherence (Figure 1).

Table 1 shows the baseline sociodemographic and lifestyle characteristics of the patients and their comorbidities and vital status recorded during follow-up according to their antihypertensive treatment adherence appropriateness. Inappropriate antihypertensive treatment adherence was significantly more frequent among younger adults, men, Roma people, current smokers, and COPD patients. Among DM patients, appropriate ATAP was more likely.

A total of 493 (9.94%) deaths occurred among participants with hypertension during the follow-up period. Patients with inappropriate ATAP experienced a higher risk of lethal outcomes (33.48%; 362 deaths among 1081 patients) than patients with appropriate ATAP (3.38%; 131 deaths among 3881 patients).

The crude mortality rate was higher in patients with inappropriate ATAP (334.8/1000) than in those who were appropriately treated (33.7/1000).

Univariate analysis revealed a significantly increased risk of lethal outcome among inappropriate ATAP patients (OR = 14.4, 95%CI: 11.6–17.8), patients aged 46–64 years (OR = 5.11, 95%CI: 3.78–6.91), current smokers (OR = 2.51, 95%CI: 2.00–3.15), former smokers (OR = 2.12, 95%CI: 1.61–2.78), DM patients (OR = 2.79, 95%CI: 2.14–3.63), IHD patients (OR = 4.17, 95%CI: 2.66–6.52), COPD patients (OR = 4.81, 95%CI: 3.36–6.88), and CA patients (OR = 3.85, 95%CI: 2.67–5.56). Female sex (OR = 0.50, 95%CI: 0.41–0.61) and high educational attainment (OR_vocational school_ = 0.67, 95%CI: 0.54–0.84; OR_graduation_ = 0.45, 95%CI: 0.35–0.58; OR_advanced_ = 0.36, 95%CI: 0.23–0.56) appeared to have a protective effect.

Multivariable logistic regression analysis confirmed the results of the univariate analyses. The significant protective factors were female sex (AOR = 0.56, 95%CI: 0.43–0.75) and high educational attainment (AOR_vocational school_ = 0.59, 95%CI: 0.43–0.82; AOR_graduation_ = 0.44, 95%CI: 0.31–0.64; AOR_advanced_ = 0.31, 95%CI: 0.17–0.56). However, inappropriate treatment adherence (AOR = 56.2, 95%CI: 41.9–75.4), older age (46–64 years; AOR = 29.8, 95%CI: 20.4–43.5) and all investigated comorbidities (AOR_DM_ = 2.61, 95%CI: 1.76–3.86; AOR_IHD_ =2.76, 95%CI: 1.47–5.18; AOR_COPD_ = 2.37, 95%CI: 1.36–4.12; AOR_CA_ = 3.28, 95%CI: 1.88–5.70) were significantly associated with an increased risk of lethal outcome. Similarly, smoking was also a risk factor (AOR_current smoker_ = 2.46, 95%CI: 1.81–3.34; AOR_former smoker_ = 1.58, 95%CI: 1.08–2.30) (Table 2).

## 4. Discussion

### 4.1. Main Findings

Our study investigated the role of nonadherence in the midterm vital prognosis of HTN among middle-aged patients. This cohort study, with 7 years of follow-up, controlled for well-known sociodemographic, lifestyle, and clinical risk factors and highlighted the prominent role of medication nonadherence. The crude mortality among HTN patients at 7 years was 334.8/1000. The risk-increasing role of inappropriate ATAP was an order of magnitude greater than those of comorbidities such as DM, IHD, COPD, and CA; smoking; male sex; and low education level. The AOR for inappropriate ATAP was extremely high (AOR = 56.2; 95%CI: 41.9–75.4).

According to a recent systematic review, both all-cause mortality (HR = 1.32; 95%CI: 1.14–1.51) and cardiovascular mortality (HR = 1.61; 95%CI: 1.43–1.781) were higher among HTN patients with low adherence [37]. Papers published after this review have confirmed low medication adherence as a risk factor of mortality, and the effect size was in the range observed in this review [38,39,40,41,42]. Our study demonstrated that in the deprived population targeted by the HES, the role of medication adherence is much stronger than usual.

Our findings related to each studied confounding factor, except for Roma ethnicity, met expectations. Because the increased risks associated with older age [43,44], male sex [45,46,47,48], low education level [49], smoking [50,51], and accompanying chronic diseases [51,52] have been demonstrated by many investigations, these results are fully consistent with mainstream knowledge. Roma ethnicity has been consistently described as a risk factor for the prognosis of cardiovascular diseases [35,53,54,55,56]. The inability of our study to demonstrate this association could be due to the small number of patients of Roma ethnicity in the sample, to the misclassification caused by the self-declaration of the ethnicity, and to the homogeneity of the population in the study area. The intervention study was intentionally conducted in a severely disadvantaged community where the lifestyles of Roma and non-Roma people differ little.

International publications are consistent with our observations of the associations between low adherence, indicated by inappropriate ATAP in our study, and younger age [56,57], male sex [56], and current smoking status [58,59]. However, the expected association between a low level of education and low adherence [45,59] was not observed in our study. Furthermore, the finding of Roma ethnicity as a risk factor for low antihypertensive treatment adherence in our investigation contradicted the findings of previous publications of increased medication adherence among Roma people [60].

### 4.2. Strengths and Limitations

We considered the most important sociodemographic and lifestyle-related confounding factors as well as the most common relevant comorbidities, thereby diminishing the impact of confounding bias. Data were collected by trained staff members at the HES and using standardized methods in the NHIF’s IT system. Misclassification was minimized by these approaches. Because the NHIF covers the entire population of Hungary, the dropout did not reduce the effectiveness of follow-up. The HES ensured highly completed records. Altogether, 4.15% of patients were excluded from the analysis because of the record incompleteness.

This study has several limitations. Some potential confounders (alcohol consumption, diet, physical activity, types of antihypertensive drugs applied and clinical severity of HTN) were not included in the data collection; and there were comorbidities (e.g.,: chronic kidney diseases) not included in the regression model. Self-reporting of ethnicity may have led to an underestimation of the number of patients of Roma ethnicity, which may have resulted in misclassification. Because the health insurance records contain only the date of death and not the cause of death, we could not separate deaths due to HTN and deaths due to other diseases in our analysis. This may have led to misclassification of the outcome. Further studies are needed to evaluate the specific impact of medication nonadherence on the cardiovascular death risk.

Because the exact dates of drug redemptions were not available for our study, we could not directly assess survival time after the discontinuation of medication. Due to this limitation of exposure assessment, we were unable to apply the Cox model.

The unusually strong association between poor adherence and mortality within 7 years observed in our study, which was conducted in an unusually disadvantaged population, warrants further study. The role of residual confounding should be clarified. Confirming the observed finding, clarifying the role of residual confounding, and exploring the mechanisms that translate extreme poverty to extremely poor adherence, thus leading to extreme mortality, are necessary.

### 4.3. Implications

The pivotal role of medication nonadherence in midterm hypertension prognosis observed in our sample can be explained by the extremely high cardiovascular medication nonadherence observed in Hungary, where 59.4% of cardiovascular prescriptions are dispensed [61]. Given that the dispensed-to-written prescription ratio (DWR) is highly variable across Hungarian GMPs, the NHIF’s primary health care monitoring system should incorporate the DWR to identify GMPs that require interventions, as recommended by the World Health Organization [62].

The direct practical conclusion from our study concerns the importance of regular patient-level checks for the redemption of prescribed medications. Fortunately, the recent development of IT-supported GMPs in Hungary makes this activity technically possible. The GP or the nurse can check whether the prescribed medicine is dispensed. Given that interventions can help patients who are not willing or able to redeem the prescribed medicine [29], this opportunity should be integrated much more rigorously into the primary health care protocol for HTN.

## 5. Conclusions

Taken together, our results revealed the following: (1) in the studied group of patients aged 18–64 years, 87.2% of the person-time of antihypertensive treatment is covered by the appropriate redemption of medications in a Hungarian, extremely disadvantaged population; (2) nonadherence to medication is more common among younger adults, men, Roma people, current smokers and COPD patients, whereas the likelihood of appropriate adherence is higher among DM patients; (3) according to the multivariable modeling, medication nonadherence is an extremely strong risk factor for a lethal outcome of HTN during the 7-year follow-up period; and (4) the method by which the GMP-level aggregate indicators can be produced in a routine manner as well as patient-level routine checks on medication nonadherence should be applied much more rigorously, as detected nonadherence should be considered a signal for necessary intervention at both the GMP and patient levels to improve the prognosis of HTN.

## Figures and Tables

**Figure 1 jcm-14-08321-f001:**
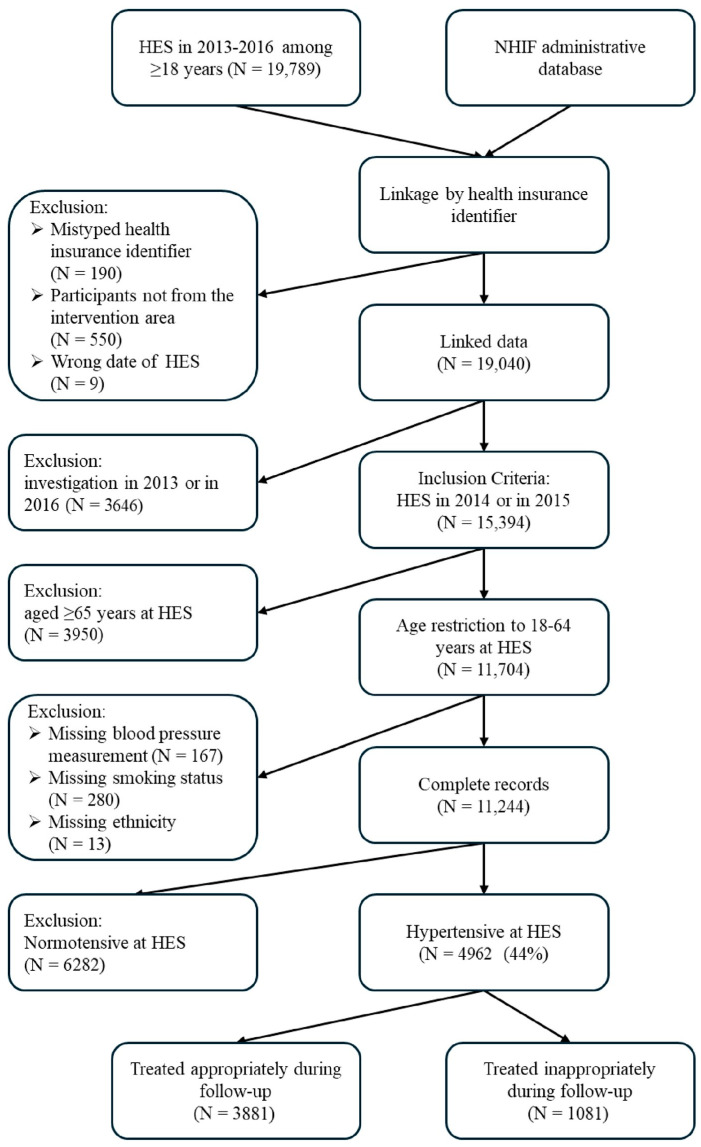
Flow diagram of patients’ cohort establishment.

**Table 1 jcm-14-08321-t001:** Distribution of the characteristics of hypertensive patients according to their antihypertensive treatment adherence appropriateness.

Variables	Categories	Appropriate Treatment Adherence N (%)	Inappropriate Treatment Adherence N (%)	Total N	*p*-Value *
Age	18–45 years 46–64 years	1019 (61%) 2862 (87%)	642 (39%) 439 (13%)	1661 3301	**<0.001**
Sex	Male Female	1762 (71%) 2119 (85%)	715 (29%) 366 (15%)	2477 2485	**<0.001**
Educational level	Eight years of school Vocational school Graduation Advanced	1259 (78%) 1267 (80%) 1017 (77%) 338 (78%)	363 (22%) 313 (20%) 312 (23%) 93 (22%)	1622 1580 1329 431	0.100
Roma ethnicity	Non-Roma Roma	3653 (79%) 228 (69%)	979 (21%) 102 (31%)	4632 330	**<0.001**
Smoking	Non-smokers Current smokers Former smokers	1654 (80%) 1456 (74%) 771 (82%)	407 (20%) 506 (26%) 168 (18%)	2061 1962 939	**<0.001**
Diabetes mellitus	No Yes	3568 (78%) 313 (82%)	1014 (22%) 67 (18%)	4582 380	**0.040**
Ischemic heart disease	No Yes	3808 (78%) 73 (77%)	1059 (22%) 22 (23%)	4867 95	0.740
COPD	No Yes	3784 (79%) 97 (66%)	1032 (21%) 49 (34%)	4816 146	**<0.001**
Cancer	No Yes	3771 (78%) 110 (73%)	1040 (22%) 41 (27%)	4811 151	0.100
Lethal outcome	Survived Died	3750 (84%) 131 (27%)	719 (16%) 362 (73%)	4469 493	**<0.001**
Total		3881	1081	**4962**	

* chi-square test.

**Table 2 jcm-14-08321-t002:** Risk factors for lethal outcomes among patients with hypertension during the 7-year follow-up period according to univariate and multivariable logistic regression models.

Variables	Categories	Lethal Outcome N (%)	Survived N (%)	Crude Mortality Rate /1000	OR [95%CI] *	aOR [95%CI] *
Antihypertensive treatment adherence appropriateness	Appropriate Inappropriate	131 (27%) 362 (73%)	3750 (84%) 719 (16%)	33.7 334.8	Ref ** **14.4 [11.6–17.8]**	**Ref** **56.2 [41.9–75.4]**
Age	18–45 years 46–64 years	49 (10%) 444 (90%)	1612 (36%) 2857 (64%)	29.5 134.5	Ref **5.11 [3.78–6.91]**	Ref **29.8 [20.4–43.5]**
Sex	Male Female	320 (65%) 173 (35%)	2157 (48%) 2312 (52%)	129.1 69.6	Ref **0.50 [0.41–0.61]**	Ref **0.56 [0.43–0.75]**
Educational level	Eight years of school Vocational school Graduation Advanced	224 (45%) 155 (32%) 90 (18%) 24 (5.0%)	1398 (31%) 1425 (32%) 1239 (28%) 407 (9.0%)	138.1 98.1 67.7 55.6	Ref **0.67 [0.54–0.84]** **0.45 [0.35–0.58]** **0.36 [0.23–0.56]**	Ref **0.59 [0.43–0.82]** **0.44 [0.31–0.64]** **0.31 [0.17–0.56]**
Roma ethnicity	Non-Roma Roma	454 (92%) 39 (8.0%)	4178 (93%) 291 (7.0%)	98.0 118.1	Ref 1.23 [0.87–1.74]	Ref 0.93 [0.57–1.51]
Smoking	Non-smokers Current smokers Former smokers	120 (24%) 264 (54%) 109 (22%)	1941 (43%) 1698 (38%) 830 (19%)	58.2 134.5 116.0	Ref **2.51 [2.00–3.15]** **2.12 [1.61–2.78]**	Ref **2.46 [1.81–3.34]** **1.58 [1.08–2.30]**
Diabetes mellitus	No Yes	411 (83%) 82 (17%)	4171 (93%) 298 (7.0%)	89.6 215.7	Ref **2.79 [2.14–3.63]**	Ref **2.61 [1.76–3.86]**
Ischemic heart disease	No Yes	464 (94%) 29 (6.0%)	4403 (98%) 66 (2.0%)	95.3 305.2	**Ref** **4.17 [2.66–6.52]**	Ref **2.76 [1.47–5.18]**
COPD	No Yes	445 (90%) 48 (10%)	4371 (98%) 98 (2.0%)	92.4 328.7	Ref **4.81 [3.36–6.88]**	Ref **2.37 [1.36–4.12]**
Cancer	No Yes	450 (91%) 43 (9.0%)	4361 (98%) 108 (2.0%)	93.5 284.7	Ref **3.85 [2.67–5.56]**	Ref **3.28 [1.88–5.70]**

* 95%CI: 95% confidence interval; ** Ref: reference category; statistically significant results are shown in **bold**; Nagelkerke r^2^ for multivariable model: 0.528.

## Data Availability

Upon reasonable request, the corresponding author will provide access to the datasets utilized or examined in this study.

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
