# Peer review of "Impact of Antihypertensive Treatment Adherence on Premature Mortality over Seven Years: A Follow-Up Investigation"

_jcm, 2025, doi:10.3390/jcm14238321_

Round 1
Reviewer 1 Report
Comments and Suggestions for Authors
Abstract
Clear objectives and results.
- The AOR (56.2) for mortality risk is unusually high-authors should discuss possible model overfitting, unmeasured confounding or data linkage artifacts even briefly in the abstract.
- Suggest including “Hungarian disadvantaged population” explicitly to contextualize the cohort.
Introduction
Background is comprehensive and up-to-date with 2025 WHO data,
Should Review
- Several sentences (lines 51–60) duplicate information on global prevalence.
- Recommendation: Condense global statistics; instead, emphasise why Hungary represents a special case (socioeconomic deprivation, Roma population, or national insurance data availability).
- Clarify how this study builds upon previous Hungarian adherence studies (Refs. 25-26) what gap does it fill?
Methods
Clear description of cohort derivation, variable definition, and follow-up.
Should Review:
- The ATAP definition (ratio of treatment-years to survival-years) is unconventional. Authors should justify this metric with references or sensitivity analyses (e.g.comparing ATAP with MPR/PDC).
- Dichotomisation at the mean (0.872) may reduce power and distort associations; consider keeping ATAP continuous or justifying the cutoff statistically (ROC, median, or quartiles).
- Clarify whether deaths within the first year were excluded (to reduce reverse causality).
- Logistic regression for a 7-year follow-up may be less informative than Cox regression; authors should explain why logistic regression was used.
- Include how missing data were handled.
- Suggest including 95% CI notation consistently (e.g. AOR=56.2; 95% CI: 41.9–75.4).
Results
Well-structured tables and clear statistical presentation.
Should Review:
- The text states “Patients with inappropriate ATAP experienced a lower risk of lethal outcomes (27%) than patients with appropriate ATAP (73%)” (line 171–172). This appears incorrectly phrased - likely reversed interpretation (nonadherence associated with higher mortality). Needs correction.
- The mortality rate (334.8/1000) vs 33.7/1000 suggests a tenfold difference, this magnitude should be interpreted cautiously and checked for computational error.
- Figure 1 should be labeled “Flow diagram” with inclusion/exclusion counts per CONSORT-style flow.
Discussion
Discussion contextualises findings and acknowledges strengths/limitations.
Should Review
- The AOR of 56.2 is exceptionally high; similar studies show HRs between 1.3–3.0. Authors must explore alternative explanations (e.g.survival bias, coding errors, data linkage accuracy, or differences in mortality ascertainment).
- Roma ethnicity interpretation conflicts with prior data (Ref. 60) should be discussed with possible underreporting or misclassification bias.
- The claim that “Misclassification was essentially prevented” (line 240) seems too strong better to write “Minimised through standardized data collection.” Suggest mentioning potential residual confounding (e.g.depression, alcohol use, BMI).
Conclusions
Clear and policy-relevant.
Should Review
- The tone (“urgent signal for immediate intervention”) could be softened for scientific neutrality. Suggest adding a sentence emphasising that future longitudinal and interventional studies are needed to confirm causality.
References and Citations
Correct and Current:
- WHO (2023, 2025), The Lancet (2021), and most MDPI and Frontiers citations are valid, current, and correctly formatted.
Issues Found:
- Reference numbering - consistent but should be cross-checked in-text for order (e.g., line 49 cites “(1)” but ref list starts with WHO 2023; should match numeric order).
- Reference 9 (WHO Global report 2025) – ensure access date (“23 Sept 2025”) is consistent with all other references (some cite July 2025).
- Reference 32 – Hungarian source link (“kollegium.aeek.hu”) may be outdated; check accessibility.
- Reference 33–38 – recent studies cited appropriately but should include DOI format uniformly.
- Reference 49 duplicates 39 (Aune et al., 2021) — likely same paper listed twice.
- Formatting inconsistencies:
- Some journal abbreviations are inconsistent (e.g., J. Hypertens vs J Hypertens).
- Some titles missing capitalization after colon.
- Check DOIs and links — a few references (20–24) use “Available from:” but no DOI-these should be added per MDPI standards.
Overall:
|
Section |
Issue |
Recommendation |
|
Abstract |
Very high AOR |
Add cautionary note; discuss potential overestimation |
|
Methods |
ATAP definition |
Justify method and cutoff statistically |
|
Methods |
Model choice |
Consider Cox regression or justify logistic |
|
Results |
Interpretation error |
Correct statement on lethal outcomes |
|
Discussion |
Extreme odds ratio |
Discuss alternative explanations |
|
Discussion |
Overconfidence in data quality |
Rephrase “prevented” → “minimized” |
|
References |
Duplicate & outdated links |
Verify refs #9, #32, #49; add DOIs |
|
Style |
Neutral tone |
Avoid overly directive language in conclusions |
Reviewer Recommendation
Major revision required before acceptance.
- The study is valuable, but methodological justification (especially the extremely high AORs), internal validity checks, and clearer linkage between findings and prior Hungarian literature are essential
- Verify all citation and references valid for the sentence
- Verify references formatted according to the citation order.
Author Response
Dear Editor and Reviewer,
Thank you very much for the careful review of our manuscript. Please find enclosed the revised version of the manuscript “Impact of antihypertensive treatment adherence on premature mortality over seven years: A follow-up investigation” by Nafisa Mhna Kmbo Elehamer, et al.
Each comment and suggestion has been considered. The corresponding changes and refinements made in the revised paper are summarized in our response after considering each of your suggestion. Answers along with the modifications we made are summarized below (Your comments/questions are in bold capitals).
Sincerely yours, Janos Sandor (on behalf of the authors)
Answers/reflections to the comments of Reviewer-1:
ABSTRACT
CLEAR OBJECTIVES AND RESULTS.
- THE AOR (56.2) FOR MORTALITY RISK IS UNUSUALLY HIGH-AUTHORS SHOULD DISCUSS POSSIBLE MODEL OVERFITTING, UNMEASURED CONFOUNDING OR DATA LINKAGE ARTIFACTS EVEN BRIEFLY IN THE ABSTRACT.
A note has been inserted to acknowledge the impact of the residual confounding.
Original text:
An extremely high mortality risk was observed among patients with inappropriate adherence (AOR= 56.2, 95%CI: 41.9–75.4), which could be attributed partly to residual confounding.
Modified text:
An extremely high mortality risk was observed among patients with inappropriate adherence (AOR= 56.2, 95%CI: 41.9–75.4), which could be attributed partly to residual confounding.
- SUGGEST INCLUDING “HUNGARIAN DISADVANTAGED POPULATION” EXPLICITLY TO CONTEXTUALIZE THE COHORT.
A note that the study focused on disadvantaged population was inserted.
Original text:
The aim of this study was to describe the prevalence of antihypertensive medication nonadherence among individuals aged 18–64 years in Hungary and its determinant factors, and to quantify the impact of antihypertensive medication nonadherence on premature mortality.
Modified text:
The aim of this study was to describe the prevalence of antihypertensive medication nonadherence among individuals aged 18–64 years in deprived Hungarian population, and its determinant factors and to quantify the impact of antihypertensive medication nonadherence on premature mortality.
INTRODUCTION
BACKGROUND IS COMPREHENSIVE AND UP-TO-DATE WITH 2025 WHO DATA,
SHOULD REVIEW
- SEVERAL SENTENCES (LINES 51–60) DUPLICATE INFORMATION ON GLOBAL PREVALENCE.
The text has been shortened as advised.
Original text:
It is considered a serious public health concern worldwide and a notable cause of premature death, making it a leading cause of health loss, with up to one in four men and one in five women experiencing this condition [2]. Over the past three decades, since 1990, the number of hypertensive people aged 30–79 years has doubled, reaching an alarming 1.3 billion individuals worldwide [3], with approximately 10 million deaths per year [4].
Modified text:
It is considered a serious public health concern worldwide and a notable cause of premature death, making it a leading cause of health loss, with up to one in four men and one in five women experiencing this condition [3-5]. Among people aged 30–79 years approximately 10 million deaths per year can be attributed to HTN [6].
- RECOMMENDATION: CONDENSE GLOBAL STATISTICS; INSTEAD, EMPHASISE WHY HUNGARY REPRESENTS A SPECIAL CASE (SOCIOECONOMIC DEPRIVATION, ROMA POPULATION, OR NATIONAL INSURANCE DATA AVAILABILITY).
- CLARIFY HOW THIS STUDY BUILDS UPON PREVIOUS HUNGARIAN ADHERENCE STUDIES (REFS. 25-26) WHAT GAP DOES IT FILL?
Together for comment-2 and comment-3:
It is demonstrated in Hungary by studies utilized the data of NHIF that the primary medication adherence is very low (59.4% of the cardiovascular prescriptions are dispensed), and the persistent use of antihypertensive medication for the single-pill combinations and free equivalent dose combinations are 82–85% and 58–73%, respectively. The relationship between these problems and the high premature cardiovascular mortality in Hungary has not been explored yet. The last sentence of the penultimate and the first sentence of the last paragraphs in “Introduction” section were modified accordingly.
Original text:
Recent Hungarian investigations have also revealed considerable challenges related to adherence [25,26].
Our study aimed to […]
Modified text:
Recent Hungarian investigations have also revealed considerable challenges related to primary medication adherence (59.4% of the cardiovascular prescriptions are dispensed [29]), and to the persistent use of antihypertensive medication for single-pill combinations and free equivalent dose combinations are 82–85% and 58–73%, respectively [30,31]. The importance of medication nonadherence in determining the poor Hungarian premature cardiovascular mortality has not yet been thoroughly investigated.
Our study based on a cohort established by an interventional study and took advantage that the cohort members’ administrative records were traceable in the administrative database of the National Health Insurance Fund (NHIF). It aimed to […]
METHODS
CLEAR DESCRIPTION OF COHORT DERIVATION, VARIABLE DEFINITION, AND FOLLOW-UP.
SHOULD REVIEW:
- THE ATAP DEFINITION (RATIO OF TREATMENT-YEARS TO SURVIVAL-YEARS) IS UNCONVENTIONAL. AUTHORS SHOULD JUSTIFY THIS METRIC WITH REFERENCES OR SENSITIVITY ANALYSES (E.G. COMPARING ATAP WITH MPR/PDC).
In reality, the ATAP is a medication possession ratio adapted to the structure of available data, which counts the lengths of periods in years. The duration of medication is approximated by the number of prescriptions (considering that one prescription covers 3 months). The total length of the observation period was quantified by the survival time for patient with lethal outcome, and with the study duration for patients alive at the end of the follow-up.
Original text:
The antihypertensive treatment adherence appropriateness (ATAP) was computed for each participant using the NHIF’s drug consumption database.
Modified text:
The antihypertensive treatment adherence appropriateness (ATAP, a medication possession ratio adapted to the structure of the available data) was computed for each participant using the NHIF’s drug consumption database.
- DICHOTOMISATION AT THE MEAN (0.872) MAY REDUCE POWER AND DISTORT ASSOCIATIONS; CONSIDER KEEPING ATAP CONTINUOUS OR JUSTIFYING THE CUT-OFF STATISTICALLY (ROC, MEDIAN, OR QUARTILES).
The ATAP as a continuous variable was not used because distribution of ATAP values was highly skewed (left-tailed).
The median ATAP was 100%. If the median were used in regression model, the perfect and non-perfect adherence would be compared – in this case the minor and major deviations from perfect adherence would be considered evenly, which would be resulted in underestimation of the nonadherence’s effect.
We used the mean as dichotomization threshold because it makes no distinction between perfect and almost perfect adherence, and this threshold ensured the high number of subjects in the two cohorts and the relatively high statistical power. (Text was not modified.)
- CLARIFY WHETHER DEATHS WITHIN THE FIRST YEAR WERE EXCLUDED (TO REDUCE REVERSE CAUSALITY).
The analysis was not controlled for the reverse causality bias. (Text was not modified.)
- LOGISTIC REGRESSION FOR A 7-YEAR FOLLOW-UP MAY BE LESS INFORMATIVE THAN COX REGRESSION; AUTHORS SHOULD EXPLAIN WHY LOGISTIC REGRESSION WAS USED.
Unfortunately, we had no free access to the whole administrative dataset of the National Health Insurance Fund. Therefore, we had no data on the dates of redemptions. (Instead, the person level totals of redemptions were available aggregated for the whole follow-up period.) The periods without redeemed antihypertensive drugs could not be determined in this setting.
Text added to the “Strength and limitations” section:
“Because the exact dates of drug redemptions were not available for our study, we could not directly assess survival time after the discontinuation of medication. Due to this limitation of exposure assessment, we were unable to apply the Cox model.”
+
When the Cox model was applied mechanistically on the two cohorts, then the Cox model could be run. See below the results:
Cox model on the survival difference between the antihypertensive treatment adherence appropriateness groups
|
Variables |
Categories |
Lethal outcome N (%) |
Survived N (%) |
Incidence Rate (per 1,000 PY) |
Crude Mortality Rate /1000 |
HR [95%CI]* |
aHRs [95%CI]* |
|
Antihypertensive treatment adherence appropriateness |
Appropriate Inappropriate |
131 (27%) 362 (73%) |
3750 (84%) 719 (16%) |
4.20 51.88 |
33.7 334.8 |
Ref** 12.7 [10.4; 15.6] |
Ref 29.2 [23.7; 36.1] |
|
Age |
18-45 years 46-64 years |
49 (10%) 444 (90%) |
1612 (36%) 2857 (64%) |
3.73 17.75 |
29.5 134.5 |
Ref 4.76 [3.54; 6.40] |
Ref 17.00 [12.4; 23.2] |
|
Sex |
Male Female |
320 (65%) 173 (35%) |
2157 (48%) 2312 (52%) |
17.27 8.82 |
129.1 69.6 |
Ref 0.50 [0.42; 0.61] |
Ref 0.58 [0.47; 0.71] |
|
Educational level |
Eight years of school Vocational school Graduation Advanced |
224 (45%) 155 (32%) 90 (18%) 24 (5.0%) |
1398 (31%) 1425 (32%) 1239 (28%) 407 (9.0%) |
18.11 12.79 8.73 7.17 |
138.1 98.1 67.7 55.6 |
Ref 0.70 [0.57; 0.86] 0.48 [0.37; 0.61] 0.39 [0.25; 0.59] |
Ref 0.68 [0.54; 0.85] 0.58 [0.45; 0.75] 0.39 [0.26; 0.61] |
|
Roma ethnicity |
Non-Roma Roma |
454 (92%) 39 (8.0%) |
4178 (93%) 291 (7.0%) |
12.73 15.71 |
98.0 118.1 |
Ref 1.22 [0.88; 1.70] |
Ref 1.01 [0.71; 1.42] |
|
Smoking |
Non-smokers Current Smokers Former Smokers |
120 (24%) 264 (54%) 109 (22%) |
1941 (43%) 1698 (38%) 830 (19%) |
7.45 17.69 15.31 |
58.2 134.5 116.0 |
Ref 2.37 [1.91; 2.95] 2.05 [1.58; 2.66] |
Ref 1.89 [1.51; 2.36] 1.37 [1.04; 1.80] |
|
Diabetes mellitus |
No Yes |
411 (83%) 82 (17%) |
4171 (93%) 298 (7.0%) |
11.63 29.31 |
89.6 215.7 |
Ref 2.53 [1.99; 3.20] |
Ref 1.69 [1.31; 2.19] |
|
Ischemic heart disease |
No Yes |
464 (94%) 29 (6.0%) |
4403 (98%) 66 (2.0%) |
12.38 43.26 |
95.3 305.2 |
Ref 3.53 [2.42; 5.14] |
Ref 1.74 [1.17; 2.59] |
|
COPD |
No Yes |
445 (90%) 48 (10%) |
4371 (98%) 98 (2.0%) |
11.97 49.92 |
92.4 328.7 |
Ref 4.13 [3.06; 5.56] |
Ref 1.48 [1.07; 2.03] |
|
Cancer |
No Yes |
450 (91%) 43 (9.0%) |
4361 (98%) 108 (2.0%) |
12.13 41.36 |
93.5 284.7 |
Ref 3.40 [2.49; 4.56] |
Ref 2.22 [1.61; 3.07] |
- INCLUDE HOW MISSING DATA WERE HANDLED.
The incomplete records had been excluded from the analysis, as it is stated in the “Methods” section (“Participants with incomplete records were excluded from our analysis.”). As it is shown in Figure 1, because of the missing blood pressure measurement (N=167), smoking status (N=280), and ethnicity (N=13), altogether 167+280+13=460 patients of 11074 were excluded. The quality of the survey could be described by this 4.15% exclusion ratio. A note was added to the “Strengths and limitations” section:
New sentence inserted:
The HES ensured highly completed records. Altogether, 4.15% of patients were excluded from the analysis because of the record incompleteness.
- SUGGEST INCLUDING 95% CI NOTATION CONSISTENTLY (E.G. AOR=56.2; 95% CI: 41.9–75.4).
Semicolons had been replaced with commas and dashes properly both in the last paragraph of Results section and in the Table 2. The “AOR=56.2, 95% CI: 41.9–75.4” formatting has been used consistently.
RESULTS
WELL-STRUCTURED TABLES AND CLEAR STATISTICAL PRESENTATION.
SHOULD REVIEW:
- THE TEXT STATES “PATIENTS WITH INAPPROPRIATE ATAP EXPERIENCED A LOWER RISK OF LETHAL OUTCOMES (27%) THAN PATIENTS WITH APPROPRIATE ATAP (73%)” (LINE 171–172). THIS APPEARS INCORRECTLY PHRASED - LIKELY REVERSED INTERPRETATION (NONADHERENCE ASSOCIATED WITH HIGHER MORTALITY). NEEDS CORRECTION.
Thanks for this comment! The numbers came from the Table 1, where all the percentages in the “Appropriate treatment adherence” and “Inappropriate treatment adherence” columns were calculated by rows. This measure is informative on the frequency of nonadherence in different strata, but is not informative about the risk of lethal outcome in the two cohorts. The phrasing in the text and the percentages derived from the Table 1 has been corrected accordingly.
Original text:
Patients with inappropriate ATAP experienced a lower risk of lethal outcomes (27%) than patients with appropriate ATAP (73%).
Modified text:
Patients with inappropriate ATAP experienced a higher risk of lethal outcomes (33.48%; 362 deaths among 1081 patients) than patients with appropriate ATAP (3.38%; 131 deaths among 3881 patients).
- THE MORTALITY RATE (334.8/1000) VS 33.7/1000 SUGGESTS A TENFOLD DIFFERENCE, THIS MAGNITUDE SHOULD BE INTERPRETED CAUTIOUSLY AND CHECKED FOR COMPUTATIONAL ERROR.
See our response in the “DISCUSSION 1” session.
- FIGURE 1 SHOULD BE LABELLED “FLOW DIAGRAM” WITH INCLUSION/EXCLUSION COUNTS PER CONSORT-STYLE FLOW.
The title of Figure 1 has been corrected.
Original title:
Figure 1. Process of patients’ cohort establishment.
Modified title:
Figure 1. Flow diagram on patients’ cohort establishment.
DISCUSSION
DISCUSSION CONTEXTUALISES FINDINGS AND ACKNOWLEDGES STRENGTHS/LIMITATIONS.
SHOULD REVIEW
- THE AOR OF 56.2 IS EXCEPTIONALLY HIGH; SIMILAR STUDIES SHOW HRS BETWEEN 1.3–3.0. AUTHORS MUST EXPLORE ALTERNATIVE EXPLANATIONS (E.G. SURVIVAL BIAS, CODING ERRORS, DATA LINKAGE ACCURACY, OR DIFFERENCES IN MORTALITY ASCERTAINMENT).
We checked many times the coding, the linkage and the statistical analysis. The extreme AOR is not a mistake. We acknowledged that the extremity reduces the convincing power. To emphasize this opinion, the last paragraph in “Strengths and limitations” section has been modified:
Original text:
The unusually strong association between poor adherence and mortality within 7 years observed in our study, which was conducted in an unusually disadvantaged population, warrants further study. Exploring the mechanisms that translate extreme poverty to extremely poor adherence, thus leading to extreme mortality, is necessary.
Modified text:
The unusually strong association between poor adherence and mortality within 7 years observed in our study, which was conducted in an unusually disadvantaged population, warrants further study. The role of residual confounding should be clarified. Confirming the observed finding, clarifying the role of residual confounding, and exploring the mechanisms that translate extreme poverty to extremely poor adherence, thus leading to extreme mortality are necessary.
- ROMA ETHNICITY INTERPRETATION CONFLICTS WITH PRIOR DATA (REF. 60) SHOULD BE DISCUSSED WITH POSSIBLE UNDERREPORTING OR MISCLASSIFICATION BIAS.
The intervention area of the study was one of the most deprived community of Hungary, as it is mentioned in the “Study design and setting” section (“…which was focused on one of the most deprived regions of Hungary …”). In this area, there is no remarkable difference between the life style of the deprived Roma and the deprived non-Roma people. Our interpretation based on this potential explanation. The “Discussion” has been modified to explicitly mention it.
Original text:
The inability of our study to demonstrate this association could be due to the small number of patients of Roma ethnicity in the sample.
Modified text:
The inability of our study to demonstrate this association could be due to the small number of patients of Roma ethnicity in the sample, to the misclassification caused by the self-declaration of the ethnicity, and to the homogeneity of the population in the study area. The intervention study was intentionally conducted in a severely disadvantaged community where the lifestyles of Roma and non-Roma people differ little.
- THE CLAIM THAT “MISCLASSIFICATION WAS ESSENTIALLY PREVENTED” (LINE 240) SEEMS TOO STRONG BETTER TO WRITE “MINIMISED THROUGH STANDARDIZED DATA COLLECTION.” SUGGEST MENTIONING POTENTIAL RESIDUAL CONFOUNDING (E.G. DEPRESSION, ALCOHOL USE, BMI).
Thank you for this comment! The text has been modified according to your suggestion.
Original text:
Misclassification was essentially prevented by these approaches.
Modified text:
Misclassification was minimised by these approaches.
CONCLUSIONS
CLEAR AND POLICY-RELEVANT.
SHOULD REVIEW
- THE TONE (“URGENT SIGNAL FOR IMMEDIATE INTERVENTION”) COULD BE SOFTENED FOR SCIENTIFIC NEUTRALITY. SUGGEST ADDING A SENTENCE EMPHASISING THAT FUTURE LONGITUDINAL AND INTERVENTIONAL STUDIES ARE NEEDED TO CONFIRM CAUSALITY.
Thank you for this comment! The text has been modified according to your suggestion.
Original text:
[…] detected nonadherence should be considered an urgent signal for immediate intervention at both the GMP and patient levels to markedly improve the prognosis of HTN.
Modified text:
[…] detected nonadherence should be considered an signal for necessary intervention at both the GMP and patient levels to improve the prognosis of HTN.
REFERENCES AND CITATIONS
CORRECT AND CURRENT:
- WHO (2023, 2025), THE LANCET (2021), AND MOST MDPI AND FRONTIERS CITATIONS ARE VALID, CURRENT, AND CORRECTLY FORMATTED.
ISSUES FOUND:
- REFERENCE NUMBERING - CONSISTENT BUT SHOULD BE CROSS-CHECKED IN-TEXT FOR ORDER (E.G., LINE 49 CITES “(1)” BUT REF LIST STARTS WITH WHO 2023; SHOULD MATCH NUMERIC ORDER).
- REFERENCE 9 (WHO GLOBAL REPORT 2025) – ENSURE ACCESS DATE (“23 SEPT 2025”) IS CONSISTENT WITH ALL OTHER REFERENCES (SOME CITE JULY 2025).
- REFERENCE 32 – HUNGARIAN SOURCE LINK (“KOLLEGIUM.AEEK.HU”) MAY BE OUTDATED; CHECK ACCESSIBILITY.
- REFERENCE 33–38 – RECENT STUDIES CITED APPROPRIATELY BUT SHOULD INCLUDE DOI FORMAT UNIFORMLY.
- REFERENCE 49 DUPLICATES 39 (AUNE ET AL., 2021) — LIKELY SAME PAPER LISTED TWICE.
- FORMATTING INCONSISTENCIES:
- SOME JOURNAL ABBREVIATIONS ARE INCONSISTENT (E.G., J. HYPERTENS VS J HYPERTENS).
- SOME TITLES MISSING CAPITALIZATION AFTER COLON.
- CHECK DOIS AND LINKS — A FEW REFERENCES (20–24) USE “AVAILABLE FROM:” BUT NO DOI-THESE SHOULD BE ADDED PER MDPI STANDARDS.
References have been restructured according to these recommendations.

Reviewer 2 Report
Comments and Suggestions for Authors
I have entusiastically reviewed the manuscript by Elehamer N et al. In this paper, authors revealed the following: (1) in the studied group of patients aged 18–64 years from a Hungarian, extremely disadvantaged population, 87.2% of the person-time was covered by the appropriate redemption of medications; (2) nonadherence to medication was more common among younger adults, men, Roma people, current smokers and COPD patients, whereas the likelihood of appropriate adherence was higher among patients with diabetes mellitus; (3) medication nonadherence is an extremely strong risk factor for a lethal outcome of HTN during the 7-year follow-up period; and (4) methods by which nonadherent patient behavior can be detected should be applied rigorously, and the detected nonadherence should be considered an urgent signal for immediate intervention to markedly improve the prognosis of HTN.
Following my suggestions:
-) Can you insert data about nephrological assess? (in example serum creatinine, eGFR, and albuminuria). In this way we can identify organ damage related with hypertension or CKD. Without the data we may underestimate the impact of renal complications.
-) The study edit a correlation between adherence and mortality. A more frequent evaluation of blood pressure values in hospital setting and at home, in my opinion may enrich the manuscript favouring early identification of at-risk patients. Plese if possible integrate the data to do a study more clinically relevant.
-) Grouping all deaths into a single outcome could mask significant differences in the impact of adherence on specific causes of mortality. Separating the causes would allow us to better understand which complications are most affected by non-adherence and to target more targeted interventions. Furthermore, this analysis would improve the clinical and epidemiological interpretation of the study, making it more useful for medical practice and prevention. Integrating data on specific causes of death would therefore enhance the quality and precision of the results.
-) I would have tried to distinguish between different causes of death. Evalluatio of one only generalized outcome may be a bias. Separating causes of death we can identify which complications are most influenced by non-adherence and edit targeted interventions
-) Why have authors excluded ancient patients?
-) What is your mind about the potential application of this model on a second cohort to verify its predictive efficacy?
-) Fig 1 COPD total n° may be to change?
-) In Tab2 verify that aOR of Antihypertensive treatment adherence appropriateness is correct (is high)
-) In the paper authors evaluate adherent/non-adherent patients without distinguishing between drug classes. Don't you think that may be a potential limitation? It's possible to understand whether some classes of drugs are associated with greater or lesser adherence? Adherence to a diuretic may have a different impact than an ACE inhibitor or a calcium channel blocker. Please, let me know your mind.
-) Evaluate to edit a Kaplan-Meier survival curve for adherence. Just because visualizing the impact of adherence on survival during study-time enforce the key message for readers.
In conclusion in this manuscript authors used data from a cohort of hypertensive individuals aged 18–64 years linked to the Health Insurance Fund’s medication purchasing data. The antihypertensive treatment adherence appropriateness (ATAP) was computed as the ratio of the observed time a patient properly treated to their observed survival time. ATAP was dichotomized by an observed mean of 0.872. Using adjusted odds ratios (AORs) from multivariate logistic regression models with 95% confidence intervals (CIs), we analyzed the factors influencing the mortality risk in 4962 participants over seven years of follow-up. Results revealed the following: (1) in the studied group of patients aged 18–64 years, 87.2% of the person-time of antihypertensive treatment is covered by the appropriate redemption of medications in a Hungarian, extremely disadvantaged population; (2) nonadherence to medication is more common among younger adults, men, Roma people, current smokers and COPD patients, whereas the likelihood of appropriate adherence is higher among DM patients; (3) according to the multivariable modeling, medication nonadherence is an extremely strong risk factor for a lethal outcome of HTN during the 7-year follow-up period; and (4) the method by which the GMP-level aggregate indicators can be produced in a routine manner as well as patient-level routine checks on medication nonadherence should be applied much more rigorously, as detected nonadherence should be considered an urgent signal for immediate intervention at both the GMP and patient levels to markedly improve the prognosis of HTN.
I congratulate myself with authors for the study and I thank Editor for the opportunity to do the review of this beautiful manuscript.
Best Regards
Author Response
Dear Editor and Reviewer,
Thank you very much for the careful review of our manuscript. Please find enclosed the revised version of the manuscript “Impact of antihypertensive treatment adherence on premature mortality over seven years: A follow-up investigation” by Nafisa Mhna Kmbo Elehamer, et al.
Each comment and suggestion has been considered. The corresponding changes and refinements made in the revised paper are summarized in our response after considering each of your suggestion. Answers along with the modifications we made are summarized below (Your comments/questions are in bold capitals).
Sincerely yours, Janos Sandor (on behalf of the authors)
Answers/reflections to the comments of Reviewer-2:
1.
CAN YOU INSERT DATA ABOUT NEPHROLOGICAL ASSESS? (IN EXAMPLE SERUM CREATININE, EGFR, AND ALBUMINURIA). IN THIS WAY WE CAN IDENTIFY ORGAN DAMAGE RELATED WITH HYPERTENSION OR CKD. WITHOUT THE DATA WE MAY UNDERESTIMATE THE IMPACT OF RENAL COMPLICATIONS.
The control for confounding factors was far from perfect in our investigation. This is explicitly acknowledged in the “Strength and limitations” section. To emphasize this more, we have expanded the section to mention that not all important comorbidities were controlled for.
Original text:
Some potential confounders (alcohol consumption, diet, physical activity, and clinical severity of HTN) were not included in the data collection.
Modified text:
Some potential confounders (alcohol consumption, diet, physical activity, types of antihypertensive drugs applied and clinical severity of HTN) were not included in the data collection; and there were comorbidities (e.g.: chronic kidney diseases) not included in the regression model.
2.
THE STUDY EDIT A CORRELATION BETWEEN ADHERENCE AND MORTALITY. A MORE FREQUENT EVALUATION OF BLOOD PRESSURE VALUES IN HOSPITAL SETTING AND AT HOME, IN MY OPINION MAY ENRICH THE MANUSCRIPT FAVOURING EARLY IDENTIFICATION OF AT-RISK PATIENTS. PLEASE IF POSSIBLE INTEGRATE THE DATA TO DO A STUDY MORE CLINICALLY RELEVANT.
Our study aimed at quantifying the prognostic role of medication nonadherence. In the consequences section, we tried to draw the conclusions: (1) monitoring the redemption rate at the GMP level is necessary to identify high-risk GMPs, and (2) taking advantage of the opportunities offered by fairly good IT support, frequent verification of redemptions is also necessary. We add a small practical note to the original text, by mentioning that this redemption checking could be managed by both GP and nurse.
Original text:
The GP can check whether the prescribed medicine is dispensed.
Modified text:
The GP or the nurse can check whether the prescribed medicine is dispensed.
3.
GROUPING ALL DEATHS INTO A SINGLE OUTCOME COULD MASK SIGNIFICANT DIFFERENCES IN THE IMPACT OF ADHERENCE ON SPECIFIC CAUSES OF MORTALITY. SEPARATING THE CAUSES WOULD ALLOW US TO BETTER UNDERSTAND WHICH COMPLICATIONS ARE MOST AFFECTED BY NON-ADHERENCE AND TO TARGET MORE TARGETED INTERVENTIONS. FURTHERMORE, THIS ANALYSIS WOULD IMPROVE THE CLINICAL AND EPIDEMIOLOGICAL INTERPRETATION OF THE STUDY, MAKING IT MORE USEFUL FOR MEDICAL PRACTICE AND PREVENTION. INTEGRATING DATA ON SPECIFIC CAUSES OF DEATH WOULD THEREFORE ENHANCE THE QUALITY AND PRECISION OF THE RESULTS.
I WOULD HAVE TRIED TO DISTINGUISH BETWEEN DIFFERENT CAUSES OF DEATH. EVALUATION OF ONE ONLY GENERALIZED OUTCOME MAY BE A BIAS. SEPARATING CAUSES OF DEATH WE CAN IDENTIFY WHICH COMPLICATIONS ARE MOST INFLUENCED BY NON-ADHERENCE AND EDIT TARGETED INTERVENTIONS
We have to admit that the outcome assessment was not properly detailed. This is explicitly acknowledged in the “Strength and limitations” section, and we explained the cause of this limitation. To put more emphasis on it, we have expanded the section to mention that investigating cause specific mortalities instead of the general mortality, more relevant results could be achieved.
Original text:
Because the health insurance records contain only the date of death and not the cause of death, we could not separate deaths due to HTN and deaths due to other diseases in our analysis. This may have led to misclassification of the outcome.
Modified text:
Because the health insurance records contain only the date of death and not the cause of death, we could not separate deaths due to HTN and deaths due to other diseases in our analysis. This may have led to misclassification of the outcome. Further studies are needed to evaluate the specific impact of medication nonadherence on the cardiovascular death risk.
4.
WHY HAVE AUTHORS EXCLUDED ANCIENT PATIENTS?
The main public health issue behind the study was that the avoidable mortality is very high in Hungary, and cardiovascular diseases among people less than 75 years of age dominate this statistical measure. We wanted to demonstrate the specific role of antihypertensive medication nonadherence in the etiology of the high avoidable mortality. By selecting people up to 64 years of age and following them for 7 years, we almost reached the age threshold used in the definition of the avoidable mortality. To explain this, we expanded the second paragraph of the “Introduction” section in the manuscript (and added the necessary references to the “References” section).
Original text:
[…] and the prevalence of uncontrolled HTN in Hungary is among the highest in Europe [11].
Modified text:
[…] and the prevalence of uncontrolled HTN in Hungary is among the highest in Europe [11]. Avoidable mortality is extremely high in Hungary compared to other countries of the European Union [12]. Although cardiovascular mortality is the main component of avoidable causes of death among adults under 75 years of age [13], we do not have epidemiological data on the extent to which individual risk factors for cardiovascular mortality in this age group contribute to the very high avoidable mortality in Hungary.
5.
WHAT IS YOUR MIND ABOUT THE POTENTIAL APPLICATION OF THIS MODEL ON A SECOND COHORT TO VERIFY ITS PREDICTIVE EFFICACY?
The unusually high relative risk estimation for the medication non-adherence observed in our study in a highly deprived population obviously requires confirmation and clarification on the underlying mechanisms. We have acknowledged it in the manuscript by completing the text of “Strengths and limitations” section.
Original text:
Exploring the mechanisms that translate extreme poverty to extremely poor adherence, thus leading to extreme mortality, is necessary.
Modified text:
The role of residual confounding should be clarified. Confirming the observed finding, clarifying the role of residual confounding, and exploring the mechanisms that translate extreme poverty to extremely poor adherence, thus leading to extreme mortality are necessary.
6.
FIG 1 COPD TOTAL N° MAY BE TO CHANGE?
Thanks for this comment! We changed the totals for COPD in the Table 1.
7.
IN TAB2 VERIFY THAT AOR OF ANTIHYPERTENSIVE TREATMENT ADHERENCE APPROPRIATENESS IS CORRECT (IS HIGH)
We have checked many times this unusual finding. The spreadsheet used in the analysis has been uploaded on to the FigShare (as it is requested by the Journal). The very high aOR is correct. (See the answer to Your comment-5.)
8.
IN THE PAPER AUTHORS EVALUATE ADHERENT/NON-ADHERENT PATIENTS WITHOUT DISTINGUISHING BETWEEN DRUG CLASSES. DON'T YOU THINK THAT MAY BE A POTENTIAL LIMITATION? IT'S POSSIBLE TO UNDERSTAND WHETHER SOME CLASSES OF DRUGS ARE ASSOCIATED WITH GREATER OR LESSER ADHERENCE? ADHERENCE TO A DIURETIC MAY HAVE A DIFFERENT IMPACT THAN AN ACE INHIBITOR OR A CALCIUM CHANNEL BLOCKER. PLEASE, LET ME KNOW YOUR MIND.
The control for confounding factors was far from perfect in our investigation. This is explicitly acknowledged in the “Strength and limitations” section. To emphasize it more, we have expanded the “Strengths and limitations” section to mention that the type of medication was not controlled for. Simply, because we had no access to data on the type of medications prescribed and redeemed.
Original text:
Some potential confounders (alcohol consumption, diet, physical activity, and clinical severity of HTN) were not included in the data collection.
Modified text:
Some potential confounders (alcohol consumption, diet, physical activity, types of antihypertensive drugs applied and clinical severity of HTN) were not included in the data collection; and there were comorbidities (e.g.: chronic kidney diseases) not included in the regression model.
9.
EVALUATE TO EDIT A KAPLAN-MEIER SURVIVAL CURVE FOR ADHERENCE. JUST BECAUSE VISUALIZING THE IMPACT OF ADHERENCE ON SURVIVAL DURING STUDY-TIME ENFORCE THE KEY MESSAGE FOR READERS.
Unfortunately, we had no free access to the whole administrative dataset of the National Health Insurance Fund. Therefore, we had no data on the dates of redemptions. (Instead, the person level totals of redemptions were available aggregated for the whole follow-up period.) The periods without redeemed antihypertensive drugs could not be determined in this setting.
Text added to the “Strength and limitations” section:
“Because the exact timing of drug discontinuations was not available for our study, we could not directly assess survival time after the discontinuation of medication. Due to this limitation of exposure assessment, we were unable to apply the Cox model.”
+
When the Cox model was applied mechanistically on the two cohorts, then the survival curves could be prepared, and the regression model could be run. See below the results:
- a) Kaplan–Meier survival curves for the antihypertensive treatment adherence appropriateness groups:
- b) Cox model on the survival difference between the antihypertensive treatment adherence appropriateness groups
|
Variables |
Categories |
Lethal outcome N (%) |
Survived N (%) |
Incidence Rate (per 1,000 PY) |
Crude Mortality Rate /1000 |
HR [95%CI]* |
aHRs [95%CI]* |
|
Antihypertensive treatment adherence appropriateness |
Appropriate Inappropriate |
131 (27%) 362 (73%) |
3750 (84%) 719 (16%) |
4.20 51.88 |
33.7 334.8 |
Ref** 12.7 [10.4; 15.6] |
Ref 29.2 [23.7; 36.1] |
|
Age |
18-45 years 46-64 years |
49 (10%) 444 (90%) |
1612 (36%) 2857 (64%) |
3.73 17.75 |
29.5 134.5 |
Ref 4.76 [3.54; 6.40] |
Ref 17.00 [12.4; 23.2] |
|
Sex |
Male Female |
320 (65%) 173 (35%) |
2157 (48%) 2312 (52%) |
17.27 8.82 |
129.1 69.6 |
Ref 0.50 [0.42; 0.61] |
Ref 0.58 [0.47; 0.71] |
|
Educational level |
Eight years of school Vocational school Graduation Advanced |
224 (45%) 155 (32%) 90 (18%) 24 (5.0%) |
1398 (31%) 1425 (32%) 1239 (28%) 407 (9.0%) |
18.11 12.79 8.73 7.17 |
138.1 98.1 67.7 55.6 |
Ref 0.70 [0.57; 0.86] 0.48 [0.37; 0.61] 0.39 [0.25; 0.59] |
Ref 0.68 [0.54; 0.85] 0.58 [0.45; 0.75] 0.39 [0.26; 0.61] |
|
Roma ethnicity |
Non-Roma Roma |
454 (92%) 39 (8.0%) |
4178 (93%) 291 (7.0%) |
12.73 15.71 |
98.0 118.1 |
Ref 1.22 [0.88; 1.70] |
Ref 1.01 [0.71; 1.42] |
|
Smoking |
Non-smokers Current Smokers Former Smokers |
120 (24%) 264 (54%) 109 (22%) |
1941 (43%) 1698 (38%) 830 (19%) |
7.45 17.69 15.31 |
58.2 134.5 116.0 |
Ref 2.37 [1.91; 2.95] 2.05 [1.58; 2.66] |
Ref 1.89 [1.51; 2.36] 1.37 [1.04; 1.80] |
|
Diabetes mellitus |
No Yes |
411 (83%) 82 (17%) |
4171 (93%) 298 (7.0%) |
11.63 29.31 |
89.6 215.7 |
Ref 2.53 [1.99; 3.20] |
Ref 1.69 [1.31; 2.19] |
|
Ischemic heart disease |
No Yes |
464 (94%) 29 (6.0%) |
4403 (98%) 66 (2.0%) |
12.38 43.26 |
95.3 305.2 |
Ref 3.53 [2.42; 5.14] |
Ref 1.74 [1.17; 2.59] |
|
COPD |
No Yes |
445 (90%) 48 (10%) |
4371 (98%) 98 (2.0%) |
11.97 49.92 |
92.4 328.7 |
Ref 4.13 [3.06; 5.56] |
Ref 1.48 [1.07; 2.03] |
|
Cancer |
No Yes |
450 (91%) 43 (9.0%) |
4361 (98%) 108 (2.0%) |
12.13 41.36 |
93.5 284.7 |
Ref 3.40 [2.49; 4.56] |
Ref 2.22 [1.61; 3.07] |

Round 2
Reviewer 1 Report
Comments and Suggestions for Authors
I have reviewed the author’s modifications. If the author has updated and cited the presentation appropriately and revised the references in order, I will accept the manuscript after minor corrections.
Author Response
Dear Editor and Reviewer,
Thank you very much for the careful second review of our manuscript. Please find enclosed the revised version of the manuscript “Impact of antihypertensive treatment adherence on premature mortality over seven years: A follow-up investigation” by Nafisa Mhna Kmbo Elehamer, et al.
Your suggestion has been considered. The corresponding changes made in the revised paper can be read below. (Your comments/questions are in bold capitals.)
Sincerely yours, Janos Sandor (on behalf of the authors)
Answer to the comment of Reviewer-1:
I HAVE REVIEWED THE AUTHOR’S MODIFICATIONS. IF THE AUTHOR HAS UPDATED AND CITED THE PRESENTATION APPROPRIATELY AND REVISED THE REFERENCES IN ORDER, I WILL ACCEPT THE MANUSCRIPT AFTER MINOR CORRECTIONS.
Thank you very much for your suggestion! We double checked the references. You can find the corrections in the tracked-changes manuscript.

Reviewer 2 Report
Comments and Suggestions for Authors
Dear authors
Thank you for your new version of this interesting topic. I hope that, after my suggestions and evaluation of other potential reviewers, the paper may be able to be published. Congratulation for your hard work to edit the manuscript. I thanK Editor for the oppotunity to do the review of your interesting paper.
Best Regards.
Author Response
Dear Editor and Reviewer,
Thank you very much for the careful second review and acceptance of our manuscript “Impact of antihypertensive treatment adherence on premature mortality over seven years: A follow-up investigation” by Nafisa Mhna Kmbo Elehamer, et al.
Sincerely yours, Janos Sandor (on behalf of the authors)
